# Investigating the Effectiveness of Using a Technological Approach on Students' Achievement in Mathematics–Case Study of a High School in a Caribbean Country

**Kendale Kashiem Dario Liburd and Hen-Yi Jen ***

Department of Industrial Engineering and Management, Yuan Ze University, Taoyuan City 320, Taiwan; kendale.liburd@gmail.com
* Correspondence: henyi@saturn.yzu.edu.tw; Tel.: +886-3-4638800

**Abstract:** It has always been a challenge for teachers to engage and motivate students to learn mathematics, due to the abstractness of some topics and the need for visual representation and technological resources. This study explores the effectiveness of using a technological approach on student achievement in mathematics, in general. A pre/post-test design was followed with a control and experimental group both learning the same topic over a 3-week period. A sample size of 35 (Experimental group = 18 and Control group = 17) high school students of 4th form level (Grade 10/Senior High) was taken with experimental group students taught using an interactive technological approach—GeoGebra software, in particular; while the control group learned the same material using the traditional approach without technology. GeoGebra is free software which can be used to teach different topics in mathematics education. Analysis of Covariance (ANCOVA) is applied in the study, and the findings shows that technology is an effective tool in teaching the topic of Coordinate Geometry concepts. It can be concluded that the student who was taught with the use of technology showed a higher level of conceptual understanding compared to the students who learned using the traditional method.

**Keywords:** technology in mathematics; dynamic geometry software; GeoGebra; mathematical education; mathematics technology and attitude scale (MTAS)





## 1. Introduction

In this exponentially changing environment and technological advancement, education systems should adapt to changes rapidly, as technology does, and advances day by day. Technology brings life to any and everything in society. It governs, more or less, everything people do and everything which surrounds them. We exist in an era where technology is used for quite literally everything, and sometimes, it seems as if there are no boundaries when it comes to what technology can do. So then, educators should ask themselves: why has technology not been incorporated into the classroom? Due to technology's prevalence in today's society, it continues to be the one thing that can fully grasp the attention of the very pupils the educators aim to teach. There is a plethora of reasons to use technology in mathematics classrooms. The advancement of computers has brought great innovation, and thus, school teachers need to be competent in using computers so that they would maximize its use in teaching and learning [1]. The National Council of Teachers of Mathematics [2] referred to technology as the essence in teaching and learning mathematics in this new era, due to the influence it has on mathematics and an enhancement tool for students' achievement. Recent literature also shows that using technology in teaching and learning is growing tremendously. The number one goal for educators and curriculum developers should be to help students discover their abilities through an interactive discovery process and steer them away from any negative beliefs or thoughts about any subject matter.

Mathematics is one of these subjects which always gets bad remarks or attitude towards. Too many times students give bad remarks towards mathematics in any nature or form. A high number of students acquire an automatic fear towards learning mathematics because of the perception given by many in society that, it is too difficult and only really smart people are great in mathematics. This is causing a tremendous problem and it is limiting the universe's ability in terms of developing more and more scientists and mathematicians or even well-rounded citizens to make the great discoveries our society needs for the needs of the future. Even though there are great scientific discoveries every day there, could be many more if the fear of mathematics is broken. To help eliminate this developing problem, "One way to do this is to introduce students to a variety of learning methods such as discovery learning. Using this type of contemporary teaching method allows for teaching concepts that are visual in nature using powerful visual tools such as dynamic geometry software" [3].

Numerous literature studies and investigations were done on the impact which technology may enhance in learning and teaching mathematics [4]; nonetheless, rarely were the investigations done, in the Caribbean region, on this technology-enhanced learning in mathematics topic. This study explores the potential of incorporating technology into the mathematics and sciences curriculum in St. Kitts and Nevis and intends to serve as a contribution in efforts to raise awareness on reforming the current system with first-hand evidence. It can also serve as one of the first studies observed in the Caribbean region in using a technological pedagogy inside secondary mathematics classrooms.

The purpose of this study is to investigate the effectiveness of teaching using technology, which includes the impacts/benefits of using a technology pedagogical approach upon students' achievements/performance, students' attitude towards learning mathematics with technology, as well as the outcomes and attitudes for secondary school students learning with digital tools in mathematics compared to students learning without the use of digital tools.

This study is organized as below: section one is the introduction, and section two reviews the literature about factors surrounding the use of technology in the classroom setting and teachers' willingness to incorporate technology in their classroom. Research methodology is in section three, and section four presents the statistical results and analysis. Section five concludes the study and provides recommendations for further research.

## 2. Literature Review

### 2.1. Technology and Its Role in Mathematics Learning and Teaching

Modern technologies can help increase collaboration and bring about more of an emphasis on practical applications of mathematics through modelling, visualization, and manipulation [5]. The National Council of Teachers of Mathematics, in 2011, stated that it is essential that teachers and students have regular access to technologies that support advanced mathematical sense-making, reasoning, problem-solving, and communication. Effective teachers optimize the potential of technology to develop students' understanding, stimulate their interest, and increase their proficiency in mathematics. When teachers use technology strategically, they can provide a much wider scope of access to mathematics, and its available resources, for all students.

The use of technology strategically inside the classroom is known to be effective and engaging for both students and teachers. Strategic use of technology in the teaching and learning of mathematics is the use of digital and physical tools by students and teachers in thoughtfully designed ways, and at carefully determined times, so that the capabilities of the technology enhance how students and educators learn, experience, communicate, and do mathematics [6]. Technology has become an increasingly important part of students' lives beyond school, and even within the classroom, it can also help increase their understanding of complex concepts or encourage collaboration among peers. Due to these benefits, current educational practice suggests that teachers implement some form of technology in their classrooms [7]. Technology plays an important role in the

development of the educational process [8]. Much of today's pedagogy is derived from technology as the forefront of the educational system as it serves as a medium in learning content and objectives efficiently, thereby improving the teaching and learning process. Much of today's pedagogy is derived from technology as the forefront of the educational system, as it serves as a medium in learning content and objectives efficiently, thereby improving the teaching and learning process.

Technology helps improve student skills in decision making, reasoning, and problem-solving. Technology can also help students to furnish their visual images of mathematical ideas, organizing and analyzing data, and can compute efficiently and accurately. It can also be used as a supporting tool for students who want to investigate in any area of mathematics, such as geometry, statistics, algebra, measurement, and number [2]. Researchers [9,10] suggest the use of instructional materials as one of resolutions to promote active learning in the classroom to improve learners' performances in geometric concepts. Instructional materials are educational tools used to improve learners' learning skills and understanding and also to increase learners' motivation levels of geometric understanding [11].

### 2.2. Dynamic Geometry Software and GeoGebra

"The family of software that can be categorized as Dynamic Geometry Software (DGS) has been considered, by many, as one of the most effective technological tools to foster conceptual understanding in mathematics education" [12]. Geometry is a study of space and shape and requires a high level of visualization abilities. Students tend to fail in this aspect because it is hard for the average pupil to visualize three-dimensional objects in a two-dimensional perspective [13]. The use of digital tools helps to enhance learning within the content of geometry [14]. Dynamic geometric software is one of many educational software that is at the disposal for teaching and learning mathematics. The use of digital tools can support skills and strategies that are highly relevant in the scientific and mathematical content area, such as real-world problem solving [15,16] or visualizing complex relationships [17,18]. Several dynamic tools enable students to learn the abstract areas of mathematics, for example, calculus, algebra, geometry, and trigonometry in an interactive explorative style [19–21]. The computer is effective as a tool for enhancing teaching and learning. With multimedia capabilities, students can visualize mathematical concepts that are difficult to imagine using traditional methods of teaching [22]. Apart from saving time from drawing work, these options help to identify invariant relations, and generalize problems and their solutions. The dynamic geometry systems offer a new approach of teaching for very difficult geometry tasks "The mutual intersecting of pyramids and prisms in axonometry," [23]. In a study, [24] showed that dynamic geometry software specifically Cabri3D has the potential to teach geometry of space analytically. The purpose of this study was to determine whether the 3-dimensional computer-supported activities designed by the dynamic software Cabri 3D for analytic geometry of space can help students have a better understanding and have a positive attitude or not. "Cabri 3D allows the user to construct and manipulate solid geometry objects in three dimensions via a 2D interface. By using Cabri 3D, three-dimensional objects, such as prisms, pyramids, cylinders, and cones can be constructed, rotated, and seen from a certain aspect on the screen and also prisms can be opened on the screen" [24].This study was done in St. Kitts and Nevis located in the Leeward Island of the Caribbean, the dynamic geometry software GeoGebra was used. It's a free, open-source dynamic geometry software for mathematics that combines teaching and learning, which offers geometry figures and algebra features in a perfect software environment that is available to operate online or downloaded at the software developer's website. It was designed to combine features of a dynamic geometry software (e.g., Cabri Geometry, Geometer's Sketchpad) and computer algebra systems (e.g., Derive, Maple) in a single, integrated, and easy-to-use system for teaching and learning mathematics [25]. This software was created by Markus Honenwater in 2002 and is now worldwide known and used in several studies that tried to study the effect of integrating it inside mathematics classrooms. GeoGebra has become a tool that can help

teachers to design effective instructional lessons that allow them to create mathematical objects and explore them visually and dynamically [26]. The software has many different components that are constantly being upgraded by the GeoGebra administrative team who also provides a large number of demo videos online for maneuvering the software.

Some researchers have explored the effectiveness of GeoGebra on the achievement of objectives in different mathematical topics. [27] discusses students' use of GeoGebra to construct a system of linear equations, according to given specifications, and manipulate the formulae in order to achieve certain conditions. A thesis [12], titled "The Impact of Teaching Mathematics with GeoGebra on the Conceptual Understanding of Limits and Continuity: the Case of Turkish Gifted and Talented Students" [22] applied a quasi-experimental study with nonequivalent control group post-test, only to observe the differences, on average, for high visual-spatial ability and low visual-spatial ability students after using GeoGebra for learning coordinate geometry.

Another study that showed good results when using GeoGebra was conducted [26]. This study was conducted in Malaysia at a secondary school that investigates the effectiveness of using GeoGebra in mathematics lessons. The objectives of this study were to measure the impact on student achievement, when using GeoGebra in Malaysia, and the benefits while identifying students' perception when using GeoGebra in learning mathematics. The students showed positive perceptions on GeoGebra software in terms of enthusiasm, confidence, and motivation. This software should be introduced to mathematics educators so that students can explore the world of mathematics in a wider way and make the students able to think critically and creatively [26]. Consequently, integrating educational technology into lessons improves academic achievements, because of appealing to more sense organs. The visuality increases the students' attention towards math lessons, which predominantly consist of an abstract concept [28]. The researchers agree that making more use of GeoGebra in math teaching is a factor in effective math teaching and permanent learning [29].

## 3. Research Method

### 3.1. Design of Experiment and the Hypotheses

Coordinate Geometry is the study of the relationships between points on the Cartesian plane. The objectives laid out by The Caribbean Examinations Council when learning this topic and were followed in this study for preparation of lesson plans are as follows below. In line with The Caribbean Examinations Council, Students with respect to Coordinate Geometry should be able to:

- determine the intercepts of the graph of linear functions; *x*-intercepts and *y*-intercepts, graphically and algebraically.
- determine the gradient of a straight line; Definition of gradient/slope.
- determine the equation of a straight line; Using: (a) the graph of the line; (b) the coordinates of two points on the line; (c) the gradient and one point on the line; (d) one point on the line or its gradient, and its relationship to another line.
- solve problems involving the gradient of parallel and perpendicular lines;
- determine from coordinates on a line segment: (a) the length; and, (b) the coordinates of the midpoint; The concept of magnitude or length, concept of midpoint.
- draw graphs of linear functions; Concept of linear function, types of linear function ($y = c$; $x = k$; $y = mx + c$; where $m$, $c$ and $k$ are real numbers).

This research used a quasi-experimental study with non-equivalent groups pre and post-test. The experiment group and the control group both received lessons using different teaching styles on the topic Coordinate Geometry with an emphasis on straight lines. After pre-test, both groups were taught on the topic mentioned above using different methods and tested after using the same Pre-Coordinate Geometry test and Pre-MTAS survey with minor adjustments to the Pre-Coordinate Geometry test. Data were then evaluated using results from pre and post-tests. The researcher prepared the lessons for the experiment group and prepared the teacher for the experiment group to use GeoGebra to teach the

concepts of straight lines. The researcher also prepared objectives for the traditional group teacher on the respective concepts.

**Hypothesis 1** (Null Hypothesis). *There is no statistical difference between the experimental group and control group on students' achievement/performance and attitude, measured by an assessment, between the students who had great visualization with GeoGebra with those who experienced ONLY the traditional teaching approach.*

**Hypothesis 2** (Alternative Hypothesis). *There is a statistical difference between the experimental group and control group on students' achievement/performance and attitude, measured by an assessment, between the students who had great visualization with GeoGebra with those who experienced ONLY traditional teaching approach.*

*3.2. Participants*

The research sample consisted of 35 student participants from the top two levels of level/form 4 of the school, namely the 4th year of student's secondary education. Of the 35 students, there were 14 males and 21 females with class average age of 15 years old. See Table 1 for participants' information.

**Table 1.** Information of Control and Experimental groups.

| Gender | Experimental Group | Control Group |
|---|---|---|
| Boys | 10 | 6 |
| Girls | 8 | 11 |
| Total | 18 | 17 |

The groups were divided by the Head and Deputy of the Mathematics Department for this school. The Head of Department is the current mathematics teacher of classroom the top class (4v1) of level/form 4. The Head of Department was also the experiment group teacher and the control group teacher (The Deputy) teachers the second level (4v2) below top class. The selection for the assignment of teacher in the designated group was decided by both class teachers. Groupings were decided based on students' "house colors." In St. Kitts and Nevis, all students are assigned to a color house upon entering whether high/secondary school or elementary school to compete among themselves in several different sporting activities too. Students who are in BLUE house from 4v1 (Top Class) and students who are in GREEN house from 4v2 (2nd Top Class) were assigned to the experimental group. The other students from respective classes (4v1 and 4v2) were automatically assigned to the Control group.

Two weeks before the intervention, the teachers from both groups distributed the concepts/topics to students that would be covered on the pretest. This was done so that participants could prepare themselves individually by using the different resources at hand. Even though students had no prior experience with the topic "coordinate geometry," there was a number of prerequisites which was taught to them over the years of the secondary education with relation to straight lines. For example, knowledge of the Cartesian Plane, perpendicular, and parallel lines were covered on the pre-test, and these are all prerequisites which students have learnt over the years prior to this study. Based on researchers, control, and experiment group teachers' knowledge and experience with teaching mathematics, there was no need to expose students to the topic before intervention, due to the fact that participants were exposed to straight lines over their secondary years. It is recommended that, to check normality of data, the population sample size should be more than 30. However, in this study, it was not possible to have a sample size of minimum 30 for each group, due to the small class sizes 4v1 and 4v2. This study could have only been carried out using those two classes at this school due to the COVID-19 restrictions and available resources at this school. Even though it may not be as accurate as a sample size of 30, normality was checked for both group and both are normally distributed with $p > 0.05$.

*3.3. Instrument*

This Coordinate Geometry Achievement Test (CGT) is consisted of 11 multiple choice questions and was administered to both the experimental group and control group. Pre-test used the original CGT, Post-test made some minor changes to a few questions from the CGT (6, 7, and 10). Changes in these questions were intended to test the conceptual knowledge on parallel and perpendicular lines, as well as how to determine the equation of a line when given a slope and a point. (See Appendices A and B for CGT Pre and Post Test respectively). Questions had one correct answer and three distracters [30]. The content validity of this study was examined by several Mathematics teachers in St. Kitts and Nevis, and all teachers have been teaching mathematics for more than 5 years. The experimental and control group teachers, who are head and assistant head of the mathematics department, at the school where this study was conducted are two mathematics teachers who have more than 20 years of teaching experiences at the secondary level.

All questions were extracted from Caribbean Secondary Education Certificate. The Caribbean Examinations Council (CXC) is a regional examining body that provides examinations for secondary and post-secondary candidates in 16 Caribbean countries. These 16 participating countries are:

Anguilla, Antigua and Barbuda, Barbados, Belize, British Virgin Islands, Cayman Islands, Dominica, Grenada, Guyana, Jamaica, Montserrat, St. Kitts and Nevis, St. Lucia, St. Vincent and the Grenadines, Trinidad and Tobago and Turks and Caicos Islands.

The Caribbean Examinations Council was established in 1972 under Agreement among the 16 participating Governments in the region to conduct such examinations as it may think appropriate and award certificates and diplomas on the results of any such examinations so conducted. The Mathematics and Technology Attitudes Scale (MTAS) questionnaire consists of 20 quick, readable items which can be administered in a short period of time. (See Appendix C for MTAS).

Table 2 shows the Quantitative Data Collection instruments. The internal consistency reliability was carried out using Cronbach Alpha after obtaining data from respondent test results, values greater than 0.60 are considered acceptable and values between 0.7 and 0.9 are considered good [31]. The results found to be 0.69 and 0.75 for pre and post MTAS respectively. This scale was developed in 2005 [32] and was used in this study to examine the effectiveness of GeoGebra software on the student attitude towards learning mathematics with technology. There are five subscales:

1. Mathematics Confidence [MC]
2. Confidence with Technology [TC]
3. Attitude to learning mathematics with technology [MT]
4. Affective Engagement [AE]
5. Behavioral Engagement [BE]

**Table 2.** Quantitative Data Collection instruments and intervention.

| Group | Pre-Test | Intervention Measure | Post-Test |
|---|---|---|---|
| Experimental Group | MTAS & CGT | Learn with GeoGebra | MTAS & CGT |
| Control Group | MTAS & CGT | Learn with Traditional method | MTAS & CGT |

"Likert-type scoring format was used for each of the subscales: MC, TC, MT and AE. Students were asked to indicate the extent of their agreement with each statement, on a five-point scale, from strongly agree to strongly disagree (scored from 5 to 1). A different, but similar, response set was used for the BE subscale. Students were asked to indicate the frequency of occurrence of different behaviors. A five-point system was again used—Nearly Always, Usually, About Half of the Time, Occasionally, Hardly Ever (scored again from 5 to 1)" [32].

*3.4. Intervention*

This study began the second month into the 2020/2021 academic school year. The experiment, control group teachers, and Information Technology technician assisted with administering the pre-test in two different computer labs. Students were given 1 h to complete both Coordinate Geometry Pre-test and the Pre-MTAS. There was no additional time needed because students had little or no knowledge on how to answer the questions. The intervention began approximately three weeks after administering the pre-tests. All lessons for the control group followed the same objectives as the treatment group however, both used different teaching methods. Lessons were prepared by researchers and approved by the Head and Deputy of the Mathematics Department, who were both the experiment and control group teacher, respectively. For the control group, all lessons were taught using the traditional approach using a very direct instruction; lecture, students think, in-class questioning, and homework assignments. The intervention was short and lasted over a 3 weeks' span with a total of 6 lessons. Each lesson was one hour long, and class met only 2 days weekly for one hour each. The duration of the class period was due to protocols of COVID-19 where all classes' meeting times were cut drastically. Below are the objectives and the lesson plans for each group.

***Lesson 1 Objectives: Prepare students for the experiment***

1. Introduction to GeoGebra, students will explore the software.
2. Students' Expectations will be laid out.
3. Make sure everyone is set for exploring GeoGebra and that everyone has access to the virtual classroom. (WhatsApp Group, GeoGebra classroom, Microsoft Teams)
4. Homework

    I.  Two videos will be posted inside the virtual classroom one will be on how to find the gradient manually (gradient formula) and the other how to find the gradient using GeoGebra.

***Lesson 2 Objectives: Finding the gradient of a straight line.***

1. Review homework video contents. Present a simple question (given two points and ask to find gradient) STUDENT
2. Demonstration on how to find the Gradient using downloaded video presentation and teacher.
3. Using GeoGebra

    - Plot two points (point will be given)
    - Draw straight line
    - Look at rise and run and what it means visually
    - Determine gradient
    - Plot two more points (student choice), draw the straight line and find the gradient.
4. Find the gradient of the line drawn using the gradient formula.
5. Review Class work
6. Homework

    I.  Video posted in Virtual classroom on how to find the midpoint of a line segment, length of a straight line
    II. Exercise on finding the gradient, midpoint, and length of a straight line

***Lesson 3 Objectives: Finding the midpoint and length of a straight line.***

1. Review homework exercise
2. Using GeoGebra identify the midpoint and the length of the distance between a pair of points
3. Plot two points, construct a line, find the gradient, midpoint, and length using GeoGebra
4. Teacher will demonstrate how to find the length and midpoint of a line with the aid of a video. (using the formulas)
5. Using the two points plot verify solutions in step 4.
6. Homework

I. Using the internet access to explore why lines are considered parallel or perpendicular. Video of Finding the equation of a line given two points. Video demonstrating how to find the equation of a straight line in GeoGebra.

*Lesson 4 Objectives: Identifying whether a pair of lines are parallel or perpendicular. Finding the equation of a straight line.*

1. Review parallel lines and perpendicular lines
2. Review the formula for the equation of a straight line using GeoGebra
3. Plot a pair of parallel lines determine the equation of the lines GeoGebra
4. Plot a pair of perpendicular line determine the equation of the lines in GeoGebra
5. Given two points find the gradient of the line and write down the gradient of the line parallel and perpendicular to that line.
6. Given the equation of a line write down the gradient of the line parallel and perpendicular.
7. Review Class Work
8. Homework

   I. Review Lessons 2–4
   II. Watch over previous videos from lessons
   III. Video will be posted in virtual classroom for review on equation of a line in slope intercept form
   IV. Review Exercise (Might be multiple choice online MUST be submitted no later than night before class)

*Lesson 5 Objectives: Review all the concepts taught in the previous lessons.*

1. Review exercise from lesson 4

   Review all concepts taught in lessons 1–4 and then administer Review Exercise 1–4.

*Lesson 6: Post Tests. Control Group Objectives*
The control group was not done with collaboration of teacher and researcher. The lessons were solely done by the control group teacher however, the objectives outlined were given by researchers, below are these objectives.
Sub-Topics to be covered under Coordinate Geometry:

- Gradient of a straight line
- Gradient of lines which are parallel or perpendicular to a given line
- Midpoint of a straight line
- Y-intercept
- Equation of a straight line

*Lesson 1 Objectives: Prepare students for the experiment*
*Lesson 2 Objectives: Finding the gradient of a straight line.*
*Lesson 3 Objectives: Finding the midpoint and length of a straight line.*
*Lesson 4 Objectives: Identifying whether a pair of lines are parallel or perpendicular. Finding the equation of a straight line.*
*Lesson 5 Objectives: Review all the concepts taught in the previous lessons and Review exercise.*
Administrator provided Review Exercise lessons 1–4.
*Lesson 6 Post Tests (Post Coordinate Geometry test and MTAS)*

*3.5. Data Analysis*

In this study, several statistical techniques were used such as the Mean, Standard Deviation, and ANCOVA analyzed in Minitab version 18. The purpose of this study was to test and compare students on their pre- and post- knowledge with respect to the intervention topic in order to give a balance comparative analysis when comparing pre- and post-results.

In order to determine whether there was a statistically significant improvement in both groups of students' conceptual understanding, analysis of covariance (ANCOVA) was used for pre-test results, as covariates, and post-test, as dependents/responses, and

were used to determine the difference between the control group and experimental group for both pre-test and post-test. This approach was done for both CGT and MTAS analysis. It was also used to determine whether there was a statistical difference between groups individual gain scores.

Gain score calculations are as follows:

$$Gain\ score = \frac{Post\ test - Pre\ test}{Max\ Score - Pre\ test} \tag{1}$$

where:

*Pre Test*: the actual raw score on pre test
*Post Test*: the actual raw score on post test

## 4. Results and Analysis

### 4.1. Coordinate Geometry Pre/Post-Tests

Table 3 presents the mean and standard deviation of each group on the Coordinate Geometry pre-test and post-test. According to the results in Table 3, it can be observed that the experimental group had a performance, which can be noted as better compared to the results (mean and standard deviation) of the control group.

**Table 3.** Descriptive statistics of experimental and control group.

| | | Pretest Scores | | Posttest Scores | |
|---|---|---|---|---|---|
| Group | N | M | SD | M | SD |
| Experimental | 18 | 7 | 5.28 | 14.22 | 5.85 |
| Control | 17 | 7.41 | 3.24 | 9.88 | 5.17 |

*Note:* M = Mean, SD = Standard Deviation, N = Group Total.

However, due to the results above, one still cannot conclude stating that the differences among both groups are significant. To further the investigation to make claim about the hypothesis, ANCOVA analysis was used to make final conclusion with regards to the CGT for pre-test and post-test among groups. Table 4 below shows the results for the differences in post-test mean scores between both experimental group and control group in the CGT.

**Table 4.** ANCOVA analysis.

| Variables | DF | Adj SS | Adj MS | F | P |
|---|---|---|---|---|---|
| Pre-Score | 1 | 45.44 | 45.44 | 1.51 | 0.23 |
| Group | 1 | 172.69 | 172.69 | 5.74 | 0.02 |
| Error | 32 | 963.43 | 30.11 | | |
| Total | 34 | 1173.54 | | | |

The ANCOVA results in Table 4 indicate that the F-value was 5.74 with a significant *p*-value of 0.02. This means that there is a significance difference in the mean scores of students who learned with technology (EG) and those who learned using traditional approach (CG) [30]. From the findings it can be said there was a statistically significant difference between the control group and experimental group, with respect to the Coordinate Geometry Post-test results, according to the findings listed in Table 4. The null hypothesis will be rejected that there was no significant difference between both groups after intervention (Post Test).

### 4.2. Mathematics Technology an Attitude Scale (MTAS) Results

The descriptive statistics of students' pre MTAS results and post MTAS results are summarized in Table 5. From the results in Table 4, it can be observed the experiment group had a much better response, in terms of their attitude, after the intervention with

respect to using technology in mathematics. In the initial stage, both groups had almost identical responses, which was the researchers' initial assumption. After the intervention, the Post MTAS scores the experimental group responses was more positive however, one cannot claim that the difference is significant.

**Table 5.** Descriptive Statistics of Students' Pre and Post MTAS Scores.

| | Pre-Test Scores | | | | Post-Test Scores | | | |
| | EG | | CG | | EG | | CG | |
| **Dependent Variables** | **M** | **SD** | **M** | **SD** | **M** | **SD** | **M** | **SD** |
|---|---|---|---|---|---|---|---|---|
| BE | 13.33 | 1.88 | 11.29 | 3.04 | 13.11 | 2.81 | 12.82 | 2.65 |
| TC | 12.72 | 1.87 | 13.18 | 2.33 | 14.56 | 2.46 | 11.77 | 3.29 |
| MC | 12.22 | 2.44 | 11.24 | 3.75 | 15.78 | 2.37 | 12.06 | 3.49 |
| AE | 12.17 | 2.94 | 12.77 | 2.39 | 14.50 | 2.94 | 11.94 | 3.31 |
| MT | 12.61 | 4.03 | 15.00 | 2.48 | 15.89 | 1.91 | 14.29 | 2.37 |
| Overall Attitude | 63.06 | 5.97 | 63.47 | 8.79 | 73.83 | 7.93 | 62.88 | 10.58 |

Note. EG = Experiment Group, CG = Control Group, M = Mean, SD = Standard Deviation.

In order to determine whether there was a statistically significant effect of the intervention on the student's attitude towards learning mathematics with technology, an *ANCOVA* analysis was conducted for MTAS. Table 6 shows the result summary of five categories for the MTAS.

**Table 6.** Analysis of Covariance (ANCOVA) Summary for MTAS.

| Variables | DF | Adj SS | Adj MS | F | P |
|---|---|---|---|---|---|
| **Behavioral Engagement (BE)** | | | | | |
| PRE BE | 1 | 0.05 | 0.05 | 0.01 | 0.94 |
| Group (Factors) | 1 | 0.49 | 0.49 | 0.06 | 0.80 |
| Error | 32 | 246.20 | 7.69 | | |
| Total | 34 | 246.97 | | | |
| **Technology Confidence (TC)** | | | | | |
| PRE TC | 1 | 0.90 | 0.90 | 0.11 | 0.75 |
| Group (Factors) | 1 | 69.00 | 69.00 | 8.04 | 0.01 |
| Error | 32 | 274.60 | 8.58 | | |
| Total | 34 | 343.60 | | | |
| **Mathematics Confidence (MC)** | | | | | |
| PRE MC | 1 | 7.47 | 7.47 | 0.85 | 0.37 |
| Group (Factors) | 1 | 127.50 | 127.50 | 14.44 | <0.01 |
| Error | 32 | 282.59 | 8.83 | | |
| Total | 34 | 410.97 | | | |
| **Affective Engagement (AE)** | | | | | |
| PRE AE | 1 | 1.99 | 1.989 | 0.2 | 0.66 |
| Group (Factors) | 1 | 54.11 | 54.11 | 5.42 | 0.03 |
| Error | 32 | 319.45 | 9.983 | | |
| Total | 34 | 378.69 | | | |
| **Learning Mathematics with Technology (MT)** | | | | | |
| PRE MT | 1 | 1.84 | 1.837 | 0.39 | 0.54 |
| Group (Factors) | 1 | 23.95 | 23.954 | 5.13 | 0.03 |
| Error | 32 | 149.47 | 4.671 | | |
| Total | 34 | 173.54 | | | |
| PRE TOTAL | 1 | 33.1 | 33.11 | 0.37 | 0.55 |
| Group (Factors) | 1 | 1037.0 | 1037 | 11.75 | <0.01 |
| Error | 32 | 2825.2 | 88.29 | | |
| Total | 34 | 3906.7 | | | |

According to Table 6, the impacts among both groups after the intervention. There were statistically significant differences between both groups in four of the MTAS sub-groups in favor of the experimental group.

- *Post TC: F = 8.04, p < 0.05*
- *Post MC: F = 14.44, p < 0.05*
- *Post AE: F = 5.42, p < 0.05*
- *Post MT: F = 5.13, p < 0.05*

PRE-TOTAL is the overall total comparison among both groups. It can be concluded, due to the results; $F = 11.75$, $p < 0.05$ and other for sub-groups listed above the researcher can reject the null hypothesis which stated there's no statistically significant difference among both groups, in terms of learning with technology. For the BE group there was no improvement in this area among both groups, researching will not reject the null hypothesis.

Table 7 shows the descriptive statistics (Mean and Standard deviation) of the Gain MTAS scores comparison among groups.

**Table 7.** Descriptive Statistics of Students' MTAS Gain Scores.

| | EG | | CG | |
|---|---|---|---|---|
| **Dependent Variable** | **M** | **SD** | **M** | **SD** |
| BE | −0.18 | 0.78 | 0.02 | 0.74 |
| TC | 0.21 | 0.48 | −0.48 | 1.15 |
| MC | 0.45 | 0.31 | 0.01 | 0.71 |
| AE | 0.14 | 0.65 | −0.26 | 0.71 |
| MT | 0.34 | 0.31 | −0.65 | 1.92 |
| EG VS CG GAIN TOT | 0.28 | 0.25 | −0.13 | 0.64 |

Note. BE: Behavioral Engagement. TC: Technology Confidence. MC: Mathematics Confidence. AE: Affective Engagement. MT: Learning Mathematics with Technology. Tot: Overall Total. CG: Control Group. EG Experiment Group. Gain scores were computed using Equation (1) as indicated above.

Further, to determine whether there's a statistical difference between the groups with regards to their Gain MTAS scores, ANCOVA analysis was done. Table 8 shows the results for MTAS Gain ANCOVA scores.

Table 8 shows the gain scores comparison among groups. Several groups showed great improvements and yielded a statistically significant difference between groups. These groups are as follows:

- *Gain TC: F = 5.53, p < 0.05*
- *Gain AE: F = 5.67, p < 0.05*
- *Gain MT: F = 4.68, p < 0.05*

Overall, even though they were not able to reject the null hypothesis for BE group and MC group, the research was still able to reject the null hypothesis overall according to their totals (Gain Tot) where $F = 6.43$, $p > 0.05$ and other 3 sub-groups listed above.

### 4.3. Coordinate Geometry Test Results and Discussions

The main purpose of this study is to investigate the effectiveness of teaching using technology; GeoGebra software and downloaded video presentations and an interactive class environment on 11th grade (4th form) students when being taught Coordinate Geometry. A secondary purpose was to investigate the impact of GeoGebra on students' attitudes towards learning mathematics with technology.

The ANCOVA results in Table 8 indicate that the *F*-value was 5.74 with a significant *p*-value of 0.02. The results here suggest that GeoGebra can be an effective tool and helpful to transform traditional environment and teaching while incorporating the traditional approach. Students who were taught Coordinate Geometry using GeoGebra software were significantly better in their achievement after comparing to the control group who were taught using the traditional approach. The findings showed that computer assisted instruction, as a supplement to traditional classroom instruction, is more effective than

traditional classroom instruction alone [22]. After further examinations were done on the lessons used for the intervention, it is clear as to why students in the experiment group outperformed their peers in the other group. The level of visualization which was done when introducing concepts was tremendous. These visualizations allowed the students to connect the abstractness of the concepts with their visuals. Students were not only given formulas and asked to calculate the midpoint or gradient or length of a straight line, they were also plotting these lines using the software and used the software as a tool to back check their solutions after using the formulas for manual calculations. In fact, I can recall the teacher from the experiment group was very impressed with the students after Lesson 1 of the intervention, which was the introduction to GeoGebra, the teacher mentioned it seems as if students had prior knowledge with the software because it was as if they knew more about maneuvering the software than he did. He mentioned students were engaged from the beginning of the lessons to the very end of each lesson. Another reason, perhaps, as to why the experiment group performed better in the Post Test can be due to the fact that lessons were taught in a dynamic learning environment, the lessons were interactive, teacher demonstrations while students practiced or explained to their lesser peers. The dynamic learning environment can enable students to act mathematically, to seek relationships between objects that would not be as intuitive with static paper and pen representations. This result can be supported by another study conducted, [33], who studied the effect of GeoGebra on students' achievement in Trigonometry. Their findings indicated that students who had learned trigonometry using GeoGebra were significantly better in their achievement compared to students who underwent the constructivist instruction.

**Table 8.** Analysis of Covariance (ANCOVA) Summary for MTAS GAIN SCORES.

| Variables | DF | Adj SS | Adj MS | F | p |
|---|---|---|---|---|---|
| **Behavioral Engagement (BE)** | 1 | 0.36 | 0.36 | 0.62 | 0.44 |
| Error | 33 | 19.13 | 0.58 | | |
| Total | 34 | 19.49 | | | |
| **Technology Confidence (TC)** | 1 | 4.22 | 4.22 | 5.53 | 0.03 |
| Error | 33 | 25.16 | 0.76 | | |
| Total | 34 | 29.38 | | | |
| **Mathematics Confidence (MC)** | 1 | 1.64 | 1.64 | 5.67 | 0.02 |
| Error | 32 | 9.25 | 0.29 | | |
| Total | 33 | 10.90 | | | |
| **Affective Engagement (AE)** | 1 | 1.39 | 1.39 | 3.01 | 0.09 |
| Error | 33 | 15.27 | 0.46 | | |
| Total | 34 | 16.66 | | | |
| **Learning Mathematics with Technology (MT)** | 1 | 8.33 | 8.33 | 4.68 | 0.04 |
| Error | 32 | 56.95 | 1.78 | | |
| Total | 33 | 65.29 | | | |
| **Experimental Group vs Control Group GAIN TOTAL** | 1 | 1.47 | 1.471 | 6.43 | 0.02 |
| Error | 33 | 7.55 | 0.229 | | |
| Total | 34 | 9.02 | | | |

### 4.4. Mathematics and Technology Attitude Scale (MTAS)

I.   Students in the experiment group results (Table 5) were better for their Post MTAS analysis (after intervention) in 4 out of 5 factors all which difference are statistically significant compared to the experimental group.

- *Post Technology Confidence[TC]: F = 8.04, p < 0.05*
- *Post Mathematics Confidence[MC]: F = 14.44, p < 0.05*

- *Post Affective Engagement[AE]: F = 5.42, p < 0.05*
- *Post Teaching and learning Mathematics with Technology[MT]: F = 5.13, p < 0.05*

II. When comparing experiment group and control gain on their gain scores average ([post-test—pretest]/[max score–pretest]) (Table 7) the experiment group outperformed the control group in 3 of the 5 factors/sub-groups of the MTAS all which yielded statistical significant differences. Three factors results are below;

- *Gain Technology Confidence[TC]: F = 5.53, p < α = 0.05*
- *Gain Affective Engagement[AE]: F = 5.67, p < α = 0.05*
- *Gain Teaching and learning Mathematics with Technology [MT]: F = 4.68, p < 0.05*

The results of this study indicated that there was a significant difference between both groups in favor of those who learned with GeoGebra. The findings showed that computer assisted instruction as a supplement to traditional classroom instruction is more effective than traditional instruction alone [22]. Some possible reasons for the positive impact of GeoGebra on the attitude of students towards learning mathematics are: serves as evidence that dynamic visualization does create means for deeper analysis of mathematics concepts [34], students in the 21st century are computer-literate and the opportunities to learn using technology support major attraction [26] digital environment motivates students in teaching and learning mathematics [35]. Learning becomes more attractive, as teachers have the opportunity to replace traditional teaching with an assistive technological approach to a great extent. Moreover, it also increases the digital literacy of students and teachers, which is a great benefit [36]. From the results analysis, it can be observed that there was a statistically significant difference between the groups in the Post MTAS overall. However, there was one category (Behavioral Engagement: BE) within this scale which showed there was no difference between the groups. This result did not affect the overall results but it stands out due to the fact that the researcher was expecting all categories to have a significant difference between them. This category focuses mainly on students' mentality towards learning mathematics. It had little, or anything, to do with learning with technology, which could be one of the reasons why there was no difference among the groups after the intervention.

The attitudes of participants towards using the computer as a support during their sessions were positive. Findings are supported by [37] literature who studied the Effect of Implementing Technology in a High School Mathematics classroom. Using technology in the classrooms can increase student engagement, increase motivation to learn, allow for better and help students not only feel more comfortable with learning mathematics but also allow for a deeper understanding of the mathematics concepts [12]. The use of dynamic geometry software, in general, increased overall student motivation, engagement, and achievement. In the study, students became more interested in their learning with the use of GeoGebra because it provided a dynamic, hands-on, and discovery learning environment [29].

## 5. Conclusions and Future Recommendations

### 5.1. Conclusions

According to the data from the Coordinate Geometry achievement pre-test, there has no statistically significant difference or any meaningful difference between the two groups. However, after the intervention there was an improvement among both groups. With regards to the Coordinate Geometry post-test results the teaching with GeoGebra method was more successful compared to the constructivist method. The results ($p = 0.02$) indicated that there was a statistically significant difference between the means of the students' scores on the post-test in favor of the GeoGebra group, which is also in line. This finding showed that using computer/technology to assist in instructions as a supplement to traditional classroom instruction can be more effective than the traditional instruction solely.

In the study, it shows that learning with technology had an impact on the students' attitude towards learning with technology. Moreover, the results from the MTAS sub-group 4, Learning Mathematics with technology (MT) yielded a statistically significant difference

($p = 0.03$) and it can be concluded that learning with technology had an impact on students' attitude. Overall, it shows that dynamic geometry software helps students to improve, and be more cooperative and actively engaged, in the lessons; in addition, teaching and learning mathematics with GeoGebra is effective. Positive impacts of making use of mathematical teaching and learning software helps to enhance students understanding and performance.

Even though the null hypothesis for the Gain scores for the Behavioral Engagement (BE) group and Affective Engagement (AE) group cannot be rejected; nonetheless the (Gain Total) shows statistical differences between the two groups. It indicates the attitude towards learning with technology is significantly more convincing than those who learned without the technology. The experimental group also showed that their conceptual knowledge for the concept of Coordinate Geometry, with relation to straight lines, were better than the students in the other group. They (EG) demonstrated that learning with technology has beneficial impacts of students' achievement and attitude and it is recommended by researchers to bridge the technological gap and education by incorporating more technology inside mathematics classrooms.

### 5.2. Recommendations

This study shows the beneficial factors of how using a technological approach could help reform and improve the education system of the current view on mathematics. In order to promote a change in the process of teaching and learning mathematics in St. Kitts and Nevis, the current constructivist method has to be modified and changed to accommodate the usage of technology inside the classroom using the available software. This study can be expanded across all schools, both primary and secondary, at different levels, especially classes of lower levels to analyze the true effectiveness of technology. This study focused only on the students' achievements and attitudes, but not on the teachers' perspectives of using technology in their classroom. To achieve an environment that facilitates technology usage in an inquiry-based, constructivist manner, a change in the pedagogical approach and the learning experience is fundamentally dependent on the actions and beliefs of teachers [38]. Teachers' perspective, attitude and knowledge towards using technology deems a good research as well. Teachers are most vital when it comes to using technology inside their classrooms, therefore, it is essential to understand both the teachers' view and students. Many teachers will only expend the effort required to integrate technology into their teaching practice when they can see that there are significant benefits in terms of learning outcomes [39]. It can also be suggested that teachers be trained on how to use technology within their lessons and the importance of doing such. Following this training, these teachers can be put under scrutiny in their classroom for a period of time using the skills from their technology training and then be reviewed.

This research was quite unique in nature for the fact that it took place in the heart of COVID-19. The method used in this research was face to face teaching; however, it can be executed remotely once all students are equipped with the necessary resources. This research was conducted successfully without being in the same geographical location and time as the experimental High School. All necessary contacts were made remotely with technological aids. It is also recommended that teachers should adapt the protocol of using technology more often inside, and outside, of their classroom. A number of available free software are on the web, which teachers can use to their advantage and not solely depend on the traditional aspect of teaching and learning. From this experiment, it was observed that students are really enthusiastic about learning with technology and have been able to see the visual aspect of the taught concept. This trend is highly recommended by authors of this research.

**Author Contributions:** K.K.D.L. is the first author and contributes in the research design, communication with teachers and administrative staffs at the High School where the experiment took place, assistance with executing the experiment, data analysis, methodology and the preparation of original draft. H.-Y.J. is the correspondent author and assists with overall support of the study. All authors have read and agreed to the published version of the manuscript.

**Funding:** This research received no external funding.

**Institutional Review Board Statement:** Not applicable.

**Informed Consent Statement:** Informed consent was obtained from all subjects involved in the study.

**Acknowledgments:** This study is partially supported by TaiwanICDF International Higher Education Scholarship Program. Much thank to Chun-Yen Chang from the Graduate Institute of Science Education and Department of Earth Sciences in National Taiwan Normal University and reviewers for constructive comments and suggestions. The authors express their gratitude to the administrative staff, teachers in the mathematics department and students in Verchilds High School so that the study becomes reality.

**Conflicts of Interest:** The authors declare there is no conflict of interest.

## Appendix A. Coordinate Geometry Pre-Test

Formulas. Calculators are recommended.

$$(line\ equation)\ y = mx + c\ \ OR\ y - y_1 = m(x - x_1)$$

$$Length\ of\ Line = \sqrt{(x_2 - x_1)^2 + (y_2 - y_1)^2}$$

$$midpt\ of\ line = \left(\frac{x_1 + x_2}{2}\right), \left(\frac{y_1 + y_2}{2}\right)$$

$$gradient\ (m) = \frac{y_2 - y_1}{x_2 - x_1}$$

1. Select the correct option for the slope and the y-intercept using the graph below.
   - (a) Slope = −1/3 y-intercept = (0,1)
   - (b) Slope = 1/3 y-intercept = (0,1)
   - (c) Slope = −3 y-intercept = (−1,0)
   - (d) Slope = −1/3 y-intercept = (1,0)

The coordinates of A and B are (3,5) and (7,1) respectively: Use this information to answer questions 2–5.

2. Midpoint of AB is:
   - (a) (10,3)
   - (b) (5,3)
   - (c) (5,6)
   - (d) (4,4)

3. Gradient of AB is:
   - (a) −1
   - (b) 1
   - (c) −4/−4
   - (d) It is impossible to find the gradient with only these two points

4. Length of the line AB is
   - (a) 6 units
   - (b) 5.60 units
   - (c) 5.66 units
   - (d) 5.67 units

5. Which one of the graphs below is an accurate reflection of the line of AB?

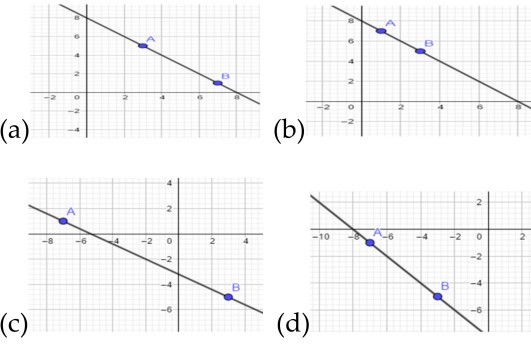

6. Two lines are said to be perpendicular if:
    (a)   They have the same gradient.
    (b)   They do not have the same gradient.
    (c)   The product of the gradient is −1 [negative 1].
    (d)   The product of the gradient is 1.

7. Two lines are said to be parallel if:
    (a)   The product of the gradient is −1 [negative 1].
    (b)   The product of the gradient is 1.
    (c)   They have the same gradient.
    (d)   They do not have the same gradient.

8. What are the gradients of the 3 lines respectively? Line A: y = 3x − 2, Line B: y = 1/2x + 5, Line C: 3x + y = 4. Select the option which has the gradient correct for all 3 line respectively. [m = gradient]
    (a)   Line A: m = 3 Line B: m = $\frac{1}{2}$ Line C: m = −3
    (b)   Line A: m = −3 Line B: m = $\frac{1}{2}$ Line C: m = 3
    (c)   Line A: m = −3 Line B: m = −$\frac{1}{2}$ Line C: m = −3
    (d)   Line A: m = 3 Line B: m = −$\frac{1}{2}$ Line C: m = −3

9. Select the correct option for the equation of the line AB from the below graph.

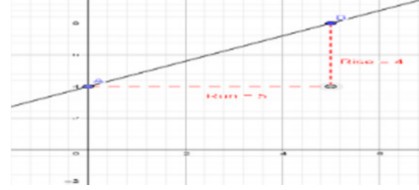

    (a)   y = x + 4
    (b)   y = 4x/5 + 4
    (c)   y = 5x/4 − 4
    (d)   y = 4x/5 − 4

10. Given two points A (−3, 5) and B (1, 1) which a straight line passes through which of the following reflects the following 3: (I). the equation of the line, (II) gradient of the line (III) the y-intercept Select the option with all the above three correct.
    (a)   equation of line AB: y = −x + 2 gradient (m) = −1 y-intercept = 2
    (b)   equation of line AB: y = −x − 2 gradient (m) = −1 y-intercept = 2
    (c)   equation of line AB: y = x − 2 gradient (m) = 1 y-intercept = −2
    (d)   equation of line AB: y = x + 2 gradient (m) = −1 y-intercept = 2

11. Complete Table of values for x + 2y = 1.

| x | −2 | −1 | 0 | 1 | 2 |
|---|----|----|---|---|---|
| y |  | 1 | 1/2 | 0 |  |

(a)     When x = −2; y = 3/2 When x = 2; y = −1/2
(b)     When x = −2; y = 2/3 When x = 2; y = −1/2
(c)     When x = −2; y = 3/2 When x = 2; y = $\frac{1}{2}$
(d)     When x = −2; y = 2/3 When x = 2; y = $\frac{1}{2}$

## Appendix B. Coordinate Geometry Post-Test

Formulas. Calculators are recommended.

$$(line\ equation)\ y = mx + c\ \ OR\ y - y_1 = m(x - x_1)$$

$$Length\ of\ Line = \sqrt{(x_2 - x_1)^2 + (y_2 - y_1)^2}$$

$$midpt\ of\ line = \left(\frac{x_1 + x_2}{2}\right), \left(\frac{y_1 + y_2}{2}\right)$$

$$gradient\ (m) = \frac{y_2 - y_1}{x_2 - x_1}$$

1.     Select the correct option for the slope and the y-intercept using the graph below.

    (a)     Slope = −1/3 y-intercept = (0,1)
    (b)     Slope = 1/3 y-intercept = (0,1)
    (c)     Slope = −3 y-intercept = (−1,0)
    (d)     Slope = −1/3 y-intercept = (1,0)

The coordinates of A and B are (3,5) and (7,1) respectively: Use this information to answer questions 2–5

2.     Midpoint of AB is:

    (a)     (10,3)
    (b)     (5,3)
    (c)     (5,6)
    (d)     (4,4)

3.     Gradient of AB is:

    (a)     −1
    (b)     1
    (c)     −4/−4
    (d)     It is impossible to find the gradient with only these two points

4.     Length of the line AB is

    (a)     6 units
    (b)     5.60 units
    (c)     5.66 units
    (d)     5.67 units

5.     Which one of the graphs below is an accurate reflection of the line of AB?

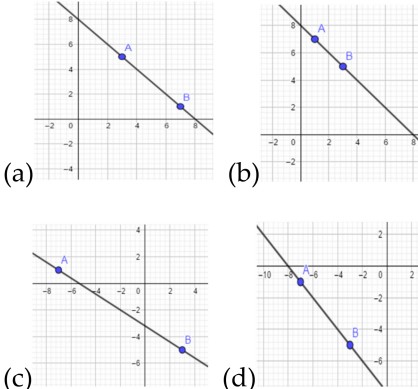

(a)  (b)  (c)  (d)

Please write/type short answers for Questions 6–7

6. Two lines are said to be perpendicular if:
7. A straight line PQ has an equation y = −5x − 3. Please select the gradient of any straight line which can be considered parallel to the line PQ.
   (a)  m = 5
   (b)  m = −5
   (c)  m = 1/5
   (d)  m = −(1/5)

8. What are the gradients of the 3 lines respectively? Line A: y = 3 x − 2, Line B: y = 1/2x + 5, Line C: 3x + y = 4. Select the option which has the gradient correct for all 3 line respectively. [m = gradient].
   (a)  Line A: m = 3 Line B: m = $\frac{1}{2}$ Line C: m = −3
   (b)  Line A: m = −3 Line B: m = $\frac{1}{2}$ Line C: m = 3
   (c)  Line A: m = −3 Line B: m = −$\frac{1}{2}$ Line C: m = −3
   (d)  Line A: m = 3 Line B: m = −$\frac{1}{2}$ Line C: m = −3

9. Using the slope intercept form [y = mx + c] of a line. The equation of the line which passes through the point (0,2) and has a gradient of 1/3 is
   (a)  y = 3x
   (b)  y = $\frac{1}{3}$x + 2
   (c)  y = 3x + 2
   (d)  y = $\frac{1}{3}$x

10. Given two points A (−3,5) and B (1,1) which a straight line passes through which of the following reflects the following 3: (I) the equation of the line (II) gradient of the line (III) the y-intercept Select the option with all the above three correct.
    (a)  equation of line AB: y = −x + 2 gradient (m) = −1 y-intercept = 2
    (b)  equation of line AB: y = −x − 2 gradient (m) = −1 y-intercept = 2
    (c)  equation of line AB: y = x − 2 gradient (m) = 1 y-intercept = −2
    (d)  equation of line AB: y = x + 2 gradient (m) = −1 y-intercept = 2

11. Complete Table of values for x + 2y = 1.

| *x* | −2 | −1 | 0 | 1 | 2 |
|---|---|---|---|---|---|
| *y* | | 1 | 1/2 | 0 | |

   (a)  When x = −2; y = 3/2 When x = 2; y = −1/2
   (b)  When x = −2; y = 2/3 When x = 2; y = −1/2
   (c)  When x = −2; y = 3/2 When x = 2; y = $\frac{1}{2}$
   (d)  When x = −2; y = 2/3 When x = 2; y = 1/2

**Appendix C.**

**Table A1.** Mathematics and Technology Attitude Scale [MTAS].

| | | | | | | |
|---|---|---|---|---|---|---|
| 1 | I concentrate hard in mathematics. | HE | Oc | Ha | U | NA |
| 2 | I try to answer questions the teacher asks. | HE | Oc | Ha | U | NA |
| 3 | If I make mistakes, I work until I have corrected them. | HE | Oc | Ha | U | NA |
| 4 | If I cannot do a problem, I keep trying different ideas. | HE | Oc | Ha | U | NA |
| 5 | I am good at using computers. | SD | D | N | A | SA |
| 6 | I am good at using things like VCRs, DVDs, MP3s and mobile phones. | SD | D | N | A | SA |
| 7 | I can fix a lot of computer problems. | SD | D | N | A | SA |
| 8 | I am quick to learn new computer software needed for school. | SD | D | N | A | SA |
| 9 | I have a mathematical mind. | SD | D | N | A | SA |
| 10 | I can get good results in mathematics. | SD | D | N | A | SA |
| 11 | I know I can handle difficulties in mathematics. | SD | D | N | A | SA |
| 12 | I am confident with mathematics. | SD | D | N | A | SA |
| 13 | I am interested to learn new things in mathematics. | SD | D | N | A | SA |
| 14 | In mathematics you get rewards for your effort. | SD | D | N | A | SA |
| 15 | Learning mathematics is enjoyable. | SD | D | N | A | SA |
| 16 | I get a sense of satisfaction when I solve mathematics problems. | SD | D | N | A | SA |
| 17 | I like using DGS for learning mathematics. | SD | D | N | A | SA |
| 18 | Using DGS is worth the extra effort | SD | D | N | A | SA |
| 19 | Mathematics is more interesting when using DGS. | SD | D | N | A | SA |
| 20 | DGS help me learn mathematics better | SD | D | N | A | SA |

HE: Hardly Ever; Oc: Occasionally; Ha: About half the time; U: Usually; NA. Nearly Always; SD: Strongly Disagree D: Disagree; N: Neutral; A: Agree; SA: Strongly Agree.

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
