# Peer review of "Investigating the Effectiveness of Using a Technological Approach on Students’ Achievement in Mathematics–Case Study of a High School in a Caribbean Country"

_sustainability, doi:10.3390/su13105586_

Round 1
Reviewer 1 Report
The paper is quite good, but for references I would like to propose using articles having a high integrative value in the current Scopus-and Web of Science-indexed literature (i.e., citing preponderantly Q1 and Q2 sources published in the past two years)
Reviewer 2 Report
I suggest the following to improve the original versione of the manuscript.
1) it would be useful to the reader providing more information about the mathematical activities the students of the experimental group carried out using Geogebra. Furthermore, it would be also useful to have more information regarding the teaching setting implemented by the teacher: teacher-centered, cooperative learning, flipped classroom etc. The same information would be useful in regard to the control group.
2) Had the students been exposed to coordinate geometry before the experimentation? If not, why was part of the pre-test about coordinate geometry, and not about students' prerequisites useful for the learning of coordinate geometry? It would be useful to discuss this point in the paper.
3) Line 16-17: while the control group learned the same material using the traditional/constructivist approach without technology.
Lines 286-288: For the control group all lessons were taught using the traditional constructivist approach using a very direct instruction; lecture, students think, in-class questioning, and homework assignments.
Lines 437-439: Learning becomes more attractive, teachers have the opportunity to replace trans-missive teaching with the constructivist method to a great extent.
It is not clear if the experimental group used a constructivist approach as a more innovative one with respect to the transmissive one, or if the control group leaped beyond a constructivist approach. What is the new approach? Does the use of Geogebra per se implies a shift in methodology and why? I could even use a transmissive approach using Geogebra.
Author Response
Please see the attachment, thank you.

Reviewer 3 Report
Review manuscript
sustainability-1193940, entitled "Investigating the effectiveness of using a technological approach on students’ achievement in Mathematics – Case Study of a High School in a Caribbean Country" SUSTAINABILITY
1 Summary of the research and your overall impression
1.1 Reviewer comment:
First of all, I would like to appreciate the efforts of the authors to carry out such a study and to try to improve the teaching of mathematics through innovative resources.
This study explores the effectiveness of using a technological approach on student 11 achievement in Mathematics in general. However, it could be highlighted as an aspect to improve that the sample was very scarce..
Below, I would like to make some specific suggestions:
2 Discussion of specific areas for improvement
2.1 Major issues
2.1.1 Reviewer comment:
Has the normality of the sample been studied?
The use of non-parametric statistics has been taken into account?
2.2 Minor issues
2.2.1 Reviewer comment:
The study presents a clear introduction with a good argument. Therefore, it is not necessary to extend it with a bibliographic review. However, the explanation and extension of the section is appreciated.
The methodology section requires revision. The first subsection talks about the design and instead the objectives of the Coordinate Geometry are explained. Furthermore, evaluation instruments are discussed when later there is a specific section where information is reiterated.
Line 222. What does (4v1) means?
Line 256. Could you explain the pilot procedure to check the validity of the instruments?
Could you briefly explain the intervention program and type of activities in each of the sessions?
It is recommended to change the value 0.00 to <0.01

Author Response
Please see the attachment, thank you.
